# Topic Modeling-Based Analysis of News Keywords Related to Patients with Diabetes during the COVID-19 Pandemic

**DOI:** 10.3390/healthcare11070957

**Published:** 2023-03-28

**Authors:** Jeong-Won Han, Jung Min Kim, Hanna Lee

**Affiliations:** 1College of Nursing Science, Kyung Hee University, Seoul 02447, Republic of Korea; 2College of Nursing Science, Kosin University, Busan 49267, Republic of Korea; 3Department of Nursing, Gangneung-Wonju National University, Gangneung-si 26403, Republic of Korea

**Keywords:** COVID-19, diabetes mellitus, news, patients, pandemics, Republic of Korea

## Abstract

This study analyzed major issues related to diabetes during the coronavirus disease (COVID-19) pandemic by using topic modeling analysis of online news articles provided by BIGKind dating from 20 January 2020, the onset of the COVID-19 outbreak in Korea, to 17 April 2022, the lifting of the social distancing restrictions. We selected 226 articles and conducted topic modeling analysis to identify the main agenda of news related to patients with diabetes in the context of the COVID-19 pandemic; both latent Dirichlet allocation and visualization were conducted by generating keywords extracted from news text as a matrix using Python 3.0. Four main topics were extracted from the news articles related to “COVID-19” and “diabetes” during the COVID-19 pandemic, including “COVID-19 high-risk group,” “health management through digital healthcare,” “risk of metabolic disease related to quarantine policy,” and “child and adolescent obesity and diabetes.” This study is significant because it uses big data related to diabetes that was reported in the mass media during the new epidemic to identify problems in the health management of patients with diabetes during a new epidemic and discuss areas that should be considered for future interventions.

## 1. Background

Following the onset of the coronavirus disease (COVID-19) in Wuhan City, China and the identification of the first confirmed case in South Korea on 20 January 2020 [1], 505,304,467 confirmed cases and 6,252,228 COVID-19 deaths have been reported worldwide, with 17,993,985 cases and 24,006 deaths reported in South Korea [2]. Chronically ill patients are more vulnerable to COVID-19, and their symptoms worsen with COVID-19 [3]. Diabetes mellitus is a representative chronic disease. Symptoms associated with hyperglycemia accelerate the aging process of each body organ and lower the immune function, thereby increasing the vulnerability and severity of infectious diseases [4]. Since the outbreak of COVID-19, patients with diabetes have been at a higher risk of viral infection and, if infected, are more likely to develop serious diseases, with an increased mortality risk [5,6]. A retrospective study in the United States revealed that COVID-19 patients suffering from diabetes exhibited more than twice the mortality rate (14.8%) of COVID-19 patients without underlying medical conditions (6.2%). The mortality rate (28.8%) of diabetic patients with poor glycemic control is nearly five times higher than that of patients without diabetes, thereby necessitating social protection and management for patients with diabetes [7].

Because anti-diabetes treatment aims to improve diabetes symptoms through strict glycemic control and maintaining normoglycemia, it is no exaggeration to state that diabetes treatment success relies on self-management [8]. Because patients must self-manage diabetes in their daily lives, it is most important that they not only receive diet and exercise therapies but also undergo examinations and receive drug prescriptions, education, and counseling for disease management [9]. With the implementation of social distancing (lockdown) measures because of the COVID-19 pandemic, activity levels have sharply decreased, and glycemic control has become more difficult as the time spent at home has increased [4]. Despite the efforts of medical institutions and public health centers to provide treatment and testing for numerous patients without taking holidays during the COVID-19 pandemic, COVID-19 has continued to spread, resulting in a significant reduction in the use of medical healthcare services. Patients suffering from chronic diseases refrain from visiting healthcare institutions [3] for fear of contracting COVID-19. This has made early diagnosis and symptomatic treatment difficult, with a resultant increase in disease severity. The limitations of COVID-19 in the overall functioning of healthcare institutions may prevent patients with diabetes from receiving timely inpatient treatment, even if they require hospitalization [9].

The COVID-19 pandemic has made it more difficult for patients with diabetes to manage their health, thus increasing the risk of personal and national disease burden. Therefore, measures should be sought that can enable patients with diabetes to treat and manage their conditions safely and actively, even during a pandemic such as COVID-19, and that can provide management and policy direction for the future. Considering that studies on patients with diabetes and COVID-19 have primarily focused on the mortality rate, it is necessary to describe the health management of diabetes during the COVID-19 pandemic to inform future diabetes management strategies. Thus, collecting related keywords, extracting topics, and comparing meanings will facilitate the understanding of the disease management status of patients with chronic diseases, including diabetes. This study uses topic modeling to better understand the types of primary concerns expressed about patients with diabetes in the media, particularly in newspapers, in the context of a new epidemic in South Korea, and to review the context where diabetes is reported to draw implications.

This study analyzed the trend of diabetes-related news articles using topic modeling and keyword network analysis while focusing on the COVID-19 epidemic period to identify the problems and implications of diabetes in the context of a health crisis.

## 2. Methods

### 2.1. Subjects

This study used news articles provided by the news big data system—BIGKinds—of the Korea Press Foundation, which were published between 20 January 2020, when the COVID-19 epidemic broke out in South Korea, and 17 April 2022, when the social distancing policy was rescinded. Nine central magazines, broadcasting media KBS, MBC, and SBS, as well as South Korea’s first news channel, YTN, were selected. The search terms for data collection included “COVID,” “Coronavirus,” “COVID-19,” “Novel infectious disease,” “Novel virus,” “Diabetes,” and “Diabetes Mellitus”. The final analysis included 226 articles excluding duplicates and articles unrelated to the search term among 1002 articles published during the search period. The Korea Press Foundation is a public institution under the Ministry of Culture, Sports, and Tourism that promotes print and online newspapers and manages the largest amount of news-related data in the Republic of Korea. BIGKinds is a big data system that was built by the Korea Press Foundation in 2016 by incorporating approximately 60 million news items gathered by 53 media outlets since 1990 into a big data analysis system.

### 2.2. Data Collection

News articles used in this study only used news published in Korean. To use news-based big data in data analysis, the data must be refined for natural language processing (NLP). BIGKinds, which uses the structural support vector machine (SVM) algorithm to automatically extract all noun keywords from the article text, remove stop words, and provide them to users as Excel files, was used in this study to perform morphological analysis and data preprocessing. The morphological analysis of news text involves extracting the morpheme, which is the smallest unit of meaning, to analyze the text. Data preprocessing involves the removal of unnecessary phrases from the selected text for analysis and normalization of terms with the same meaning but with various expressions. Morphological analysis and data preprocessing were applied to the article, after which the text found in the keyword search was extracted and used as an Excel file. For reference, the structural SVM algorithm is a machine learning algorithm for NLP of text that can recognize and classify Korean predicates with a performance of 97.13% [10].

### 2.3. Analytical Methods

In this study, topic modeling analysis was conducted to identify the main agenda of news related to patients with diabetes in the context of the COVID-19 pandemic in South Korean society, and the latent Dirichlet allocation (LDA) analysis and visualization were conducted by generating keywords extracted as a matrix from the news text using Python 3.0.

Text data collected using big data undergo data preprocessing before conversion into a format that is suitable for analysis. In this study, the keywords in the form of nouns extracted from BIGKinds were extracted from characters other than Korean characters, such as Chinese characters, numbers, and English, using the Natural Language Toolkit, Nltk, in Python 3.0. Keywords that were irrelevant to the research content were excluded by additional stop-word processing. For keyword analysis, the Scikit-learn package was used to create a document-term matrix, and term frequency analysis was performed to analyze keywords with a high frequency of occurrence; these terms were listed in order of frequency using two criteria: term frequency (TF) and term frequency-inverse document frequency (TF–IDF), which is the most commonly used weight in text-mining analysis and involves a statistical numerical representation of how important a word is within a specific document. TF–IDF is used to identify terms that, despite their high frequency of appearance in most documents, do not adequately describe the document characteristics. However, a larger weight value and higher ratio of the subject words indicate a higher importance of the term [11]. The pandas software library was used for processing and storing the data that was used in preprocessing and analytical processing. NumPy was used for mathematical calculations and algorithm construction. For visualization of the analytical results, Matplotlib was used. Topic modeling analysis based on the LDA algorithm was used to understand the structural characteristics of topics related to diabetes [11]. The advantage of the topic modeling technique is that it derives a latent topic nested in a corpus based on the relevance of the term used in the document. Gensim, a Python package, was used as a tool to measure the coherence of the LDA topic model, and pyLDAvis was used for LDA visualization. The LDA technique is used to determine whether the number of topics is clearly classified in an inter-distance map (IDM), which is a map that spatially represents the distance between topics. In the IDM, each topic appears in circles, where independent circles that do not overlap represent well-classified topics. Various numbers of topics were set to calculate the optimal number of topics (k) for the topic of the keyword. By confirming the visual change in IDM, four distinct topics were identified.

### 2.4. Research Integrity

A team of two nursing professors that comprised one nursing informatics major and one specialist treating diabetes discussed the validity of the research, set the scope of data collection, and analyzed and interpreted aspects of the data. This included the number of topics and topic names to improve the validity of the design, data collection, analysis, and interpretation of the study’s results. The parts that required correction were supplemented and analyzed, and the results were reconfirmed.

## 3. Results

### 3.1. Keyword Analysis

In this study, the top 25 keywords based on the collected TF values were reviewed. Out of a total of 1002 articles collected covering the search period, 226 articles were included in the final analysis after excluding those with data that were unrelated to the search terms; the total number of keywords used for analysis after purification among the total collected keywords was 61,351. For the TF values in the final keywords, the TF value for COVID-19 (539) was the highest, followed by chronic disease (440), health (209), vaccine (186), and hospital (178). Among the 25 keywords, the TF-IDF was the highest for social isolation (1.38), followed by hospital (1.31), care (1.12), symptom (1.12), and support (1.04) (Table 1).

### 3.2. Topic Modeling Analysis

The topic modeling in this study demonstrated that the main topics of news articles related to “COVID-19” and “diabetes” during the COVID-19 pandemic included “COVID-19 high-risk group,” “health management through digital healthcare,” “risk of metabolic disease related to quarantine policy,” and “child and adolescent obesity and diabetes.”

The main keywords in the first topic, “COVID-19 high-risk group,” included confirmed cases, underlying diseases, infectious diseases, smokers, and high-risk groups. Patients confirmed positive for COVID-19 or with severe COVID-19 and with underlying medical conditions including diabetes or smoking were frequently reported in the media as constituting high-risk groups for COVID-19, with various reports of type 2 diabetes as a sequela of COVID-19 infection (Table 2, Figure 1).


*“A South Korean research team observed why people with strokes, diabetes, or smokers were vulnerable to the new coronavirus disease (COVID-19). The National Institute of Health evaluated, ‘It is meaningful in that it has revealed the reasons why people with underlying diseases such as diabetes and strokes, are considered high-risk groups for COVID-19, and smokers are more vulnerable to COVID-19’.”*
(Yunhap News, 20 June 2022)


*“A study published in the medical journal, the Lancet Diabetes & Endocrinology, explains that a 40% increase in the risk of diabetes among confirmed COVID-19 patients implies that one in 100 cured people will be diagnosed with diabetes.”*
(JoongAng Ilbo, 23 March 2022)

The main keywords in the second topic, “health management through digital healthcare,” included dead, pregnant women, possibility, patients, healthcare workers, infectious diseases, software, and smartphones. This implies that, during the COVID-19 pandemic, patients with diabetes or hypertension and pregnant women concerned about childbirth frequently avoided hospitalization and drug prescriptions because of the risk of infection, which resulted in problems involving disease management and unmet healthcare needs. Disease management for digital healthcare was reported as a countermeasure to these social phenomena (Table 2, Figure 2).


*“One in six patients with hypertension and diabetes was observed to be passive in seeking hospital treatment due to anxiety about being infected with COVID-19. They received treatment for the chronic diseases, however, postponed the visit to hospitals for diagnosis and treatment of other diseases such as complications. This suggested that it was necessary to resume the chronic disease management program and make efforts to lower anxiety about COVID-19 infection.”*
(The Hankyoreh, 13 April 2022)


*“Due to the COVID-19 pandemic, non-face-to-face treatment is temporarily allowed for patients with COVID-19 and, so far, the number of non-face-to-face treatment cases is approaching 4 million. Therefore, it is now time to find a reasonable alternative to how telemedicine can contribute to the national economy while maintaining the public nature of healthcare. The president-elect has also expressed his intention to expand telemedicine projects and foster the digital healthcare industry.”*
(JoongAng Ilbo, 31 March 2022)


*“The novel coronavirus infection (COVID-19) has changed various things. There have been many situations where people cannot go to hospitals and pharmacies even if they want to. It is the non-face-to-face treatment platform that has filled this medical gap. Dr. Now has been used by 3.1 million people since its service started in December 2020. It is a service that allows you to receive medical treatment over the phone by installing an app on your mobile phone and selecting a symptom, and after receiving treatment, you can receive a mobile prescription and have your medicine delivered.”*
(The Dong-a Ilbo, 30 March 2022)

The main keywords in the third topic, “risk of metabolic disease related to quarantine policy,” included hypertension, severe disease, chronic disease, fatty liver, cholesterol, and being treated for COVID-19 at home. In the early stages of COVID-19, several articles reported that the high risk of infection or death from chronic diseases, including diabetes, was associated with cholesterolemia and fatty liver. Various articles reported that lifestyle changes, such as eating and drinking alone at night, following social distancing, or being treated for COVID-19 at home because of COVID-19, could cause fatty liver or elevated cholesterol, thereby increasing the risk of metabolic diseases such as obesity and diabetes (Table 2, Figure 3).


*“The reason why more people with diabetes or heart diseases die from the novel coronavirus infection (COVID-19) has been revealed. The coronavirus has been observed to jump on cholesterol molecules and easily penetrate cells. Scientists believe that blocking the binding of the virus to cholesterol could cure chronically ill patients infected with COVID-19.”*
(Chosun Ilbo, 27 November 2020)


*“Hypertension, diabetes, and dyslipidemia are items that patients tend to neglect in their health examinations. According to the results of a survey conducted by the Korean Society for Obesity in May, 46% of those who gained 3 kg or more weight after the COVID-19 epidemic increased, and the proportion of those who did not exercise increased from 18% before the COVID-19 epidemic to 32%. Weight gain increases the risk of hypertension, diabetes, and dyslipidemia. Obese people should keep an eye on their blood pressure, blood sugar, and cholesterol levels as they are likely to suffer from hypertension, diabetes, and dyslipidemia. In addition, as liver function abnormalities due to fatty liver may occur, the level of liver enzymes should be checked.”*
(Segye Ilbo, 31 October 2021)

The main keywords in the fourth topic, “child and adolescent obesity and diabetes,” included public health centers, monitoring, depression, adolescents, obesity rate, and weight. Several articles reported an increase in obesity, including overweight children and adolescents, during the COVID-19 pandemic. They emphasized the risk of lower physical activity and altered eating habits because of social distancing and other factors that lead to obesity and diabetes, as well as the increasing prevalence of depression in children and adolescents (Table 2, Figure 4).


*“As a result of checking the weight change for school-age children and adolescents due to COVID-19, all obesity-related indicators such as weight and body mass index (BMI, weight divided by the square of height) increased compared to before the closure of schools. The complication risks related to obesity, such as metabolic syndrome, fatty liver, and diabetes, also increased significantly. This may have been due to the maintenance of usual lifestyles, such as eating habits, despite the significantly decreased outside activities after the closure of schools. In particular, it is pointed out that people diagnosed with nonalcoholic fatty liver along with obesity must be cautious with blood sugar control through professional treatment.”*
(Kukmin Ilbo, 12 April 2021)


*“A US research team analyzed that obese children or those suffering from chronic diseases are more likely to develop more severe symptoms than general children and adolescents when they contract the novel coronavirus infection (COVID-19). This is the result of a large-scale study on 167,262 children and adolescents under the age of 19, and this phenomenon is noteworthy because it is the same as that observed in adults.”*
(The Dong-a Ilbo, 9 February 2022)


*“The number of obese 6–17-year-olds in China has reached 53 million, doubling in the past decade. According to a report on child obesity in China, the obesity rate among children and adolescents is expected to surge from 15% in 2020 to 28% in 2030. In addition, according to a sample survey of elementary, middle, and high schools in nine regions across China by the Ministry of Education in China, the rate of myopia increased by 11.7% between January and June of last year. According to the World Health Organization (WHO), in 2018, the rate of myopia among adolescents in China was 53.6%, the highest in the world. As the physical balance is disrupted, there is a greater concern about the mental health of adolescents than all else. According to the national study of mental health in China in 2020, 24.7% of adolescents reported depression (severe depression in 7.4%).”*
(Hankook Ilbo, 9 May 2021)

## 4. Discussion

This study was conducted through topic modeling by collecting articles related to diabetes from 20 January 2020, the date of the COVID-19 outbreak, to 17 April 2022, when the social distancing policy was rescinded in South Korea. The aim was to identify social issues and implications for diabetes management after COVID-19. The implications of the study results are as follows.

The first topic was “COVID-19 high-risk group,” which involved identifying people with diabetes, as well as those with other chronic diseases, as a vulnerable group for COVID-19. COVID-19 patients may develop type 2 diabetes as a complication, suggesting that diabetes and infectious diseases must be treated together. COVID-19 is a risk to people with diabetes because diabetes impairs the immune system, making the body vulnerable to infectious diseases [5,6]. According to the Korean Diabetes Association, 14.5–21.8% of COVID-19 patients in Korea are diabetic [14]. According to an analysis of 5000 COVID-19 patients in South Korea, the number of COVID-19 patients with diabetes who required mechanical ventilation was 1.93 times higher, and the mortality rate was 2.66 times higher. Patients with diabetes who are treated with insulin have a 25% higher risk of infection [14,15]. Controlling blood sugar levels before hospitalization in COVID-19 patients with diabetes modulates disease severity and mortality. Mortality increased when the glycated hemoglobin (HbA1c) was 7.5% or higher in the 15 months preceding hospitalization [16]. This implies that glycemic control has a significant impact on controlling infection-related symptoms. In the context of the COVID-19 pandemic, the health management vulnerability of patients with diabetes has increased as the temporary disruption of healthcare institutions has prevented them from receiving adequate care or social and economic support [17,18]. Therefore, in preparation for a future novel infectious disease condition, the government, society, and medical institutions consider it necessary to evaluate aspects that can enhance the self-care capacity in the management program for chronic diseases, such as diabetes, where various programs can link home and healthcare institutions.

The second topic was “health management through digital healthcare,” and during the COVID-19 period, media reports emphasized diabetes patient management through digital healthcare in the Republic of Korea and that more research and policy modifications are required. During COVID-19, patients were fearful of contracting COVID-19 during in-person clinic/hospital visits, leading to poor or delayed disease management. Telemedicine is a good alternative for managing diseases. In the Republic of Korea, there has been much debate about telemedicine [19,20,21]. Positive aspects, such as improved access to medical care, improved medical quality, and economic feasibility have been presented alongside issues of medical safety and effectiveness verification, accountability, collapse of the medical delivery system, and differences in healthcare and industry perspectives [22,23]. Because of COVID-19, the industrial structure has rapidly changed because of the non-face-to-face mode of management, and the WHO has issued a guide for telemedicine implementation in response to COVID-19 [24]. In South Korea, non-face-to-face treatment was temporarily permitted during the COVID-19 pandemic, which allowed patients with chronic diseases, including diabetes, to receive prescriptions via telemedicine. Furthermore, with the implementation of telemedicine, digital healthcare technology has been rapidly developing. In a study among patients with diabetes in the United States [25], when a mobile app, remote lifestyle coaching, connected devices, live video consultations, and continuous glucose monitoring (CGM) were provided, the dietary habits of patients with diabetes improved, and their HbA1c levels decreased from 7.7% to 7.1%. The importance of CGM, which enables glycemic monitoring and result transmission to a smartphone, as a new paradigm for managing patients with diabetes has recently been highlighted. Moreover, the American Diabetes Association [26] recommended CGM in its Standards of Medical Care in Diabetes recommendations, while also requiring a policy to support the use of digital healthcare devices by patients with diabetes. With the expansion of digital healthcare from wellness to disease prevention and management and the increase in public awareness and use of non-face-to-face medical care because of the COVID-19 pandemic, it is necessary to expand the scope of support and application areas for chronic diseases such diabetes. Therefore, it is necessary to discuss the area where telemedicine will be introduced in the future, rather than focusing on whether to allow telemedicine.

The third topic was the “risk of metabolic disease related to quarantine policy.” The national quarantine policy because of the outbreak of COVID-19 induced changes in people’s lifestyles, increasing the risk of chronic diseases, including diabetes, which requires countermeasures. A study conducted in the UK and Latin America found less physical activity in people with chronic conditions, such as obesity, after the COVID-19 outbreak [27,28]. According to the results of a survey on “changes in health behavior before and after the COVID-19 pandemic” among 1500 men and women aged 20–65 years in South Korea, only 35.9% of men and 29.1% of women responded that they were physically active, while those who stopped engaging in physical activity comprised 48.7% of men and 47.0% of women. Furthermore, because of the COVID-19 outbreak, the time spent at home, such as working from home, has increased, followed by an increase in food deliveries, which interfere with the dietary adherence of patients with diabetes [29]. For patients with diabetes, exercise and dietary management are essential factors for lowering blood sugar. Individuals with chronic diseases, including diabetes, should see a doctor regularly, check their health status, and receive medication. Nevertheless, these individuals are classified as a “high-risk group for COVID-19” and are discouraged from going out or visiting medical institutions. Therefore, they experience difficulties in managing chronic diseases because healthcare is focused on the prevention of COVID-19. Chronic diseases, such as obesity, hypertension, and diabetes, are frequently asymptomatic in the early stages and are called “silent killers.” During the COVID-19 pandemic, chronic diseases of people without proper treatment and steady management may worsen [30]. Therefore, when the national anti-epidemic policy is implemented in response to a new infectious disease, the government and local communities should provide education and publicity programs that can prevent metabolic diseases from worsening because of changes in physical activity and eating habits that are associated with chronic diseases such as diabetes mellitus. A system that can periodically check and provide feedback is required, in addition to professional workforce management.

The fourth topic is “child and adolescent obesity and diabetes.” During the COVID-19 period, the management of obesity and diabetes in children and adolescents has emerged as an important keyword in South Korea and many countries worldwide, and the need for management has been emphasized. According to a study conducted in the UK, the number of cases of diabetes in children and adolescents increased to twice that of the previous year during the COVID-19 epidemic [31]. In the United States, children infected with COVID-19 were 2.67 times more likely to develop diabetes [32], indicating the urgent need to manage diabetes in children and adolescents after the COVID-19 pandemic. During the COVID-19 pandemic, the risk of diabetes increased as outdoor activities were reduced, with children and adolescents spending more time indoors, resulting in weight gain. According to a previous study [33], the proportion of overweight or obese children was 25.3% for girls and 23.3% for boys among 113 patients who visited the hospital from May to July 2019 before the outbreak of COVID-19. An analysis of 201 children who visited the hospital one year later showed that the proportion of overweight or obese children was 31.4% for girls and 45.8% for boys, indicating a significant increase. In particular, the obesity rate increased by 6.1% for girls and 22.5% for boys in 1 year, and the high obesity rate doubled. Hyperlipidemia, fatty liver, hypertension, and diabetes require proper health management. Since 2006, the WHO [34] has adopted the “Charter on Counteracting Obesity” and has proposed measures to regulate food advertisements for children, promote low-salt, low-sugar, and low-fat processed foods, and strengthen the management of school nutrition and physical education. The risk of childhood and adolescent obesity has long been a global warning signal, but the severity has increased following the outbreak of COVID-19. Therefore, future diabetes management should focus on the management of obesity and diabetes in children and adolescents and implement policies that necessitate a link between schools and childcare institutions for obesity and diabetes management.

Topic 3 and Topic 4 had many overlapping parts, but the commonality of the two topics was that social isolation led to changes in health behavior, such as physical inactivity, contributing to weight gain and obesity. Obesity exacerbates the risk of metabolic syndrome and increases the odds of having a severe outcome from COVID-19. However, the biggest difference between Topic 3 and 4 was that the subjects of Topic 3 were primarily adults, and the subjects of Topic 4 were mostly children and adolescents. Therefore, to prevent and manage diabetes, appropriate policies and interventions are needed depending on the subject. Particularly, preventing childhood obesity is extremely important as it plays a protective role in preventing the onset of type-two diabetes as adults.

## 5. Conclusions

This implies that diabetes and infectious diseases must be treated together. During the COVID-19 period, remote treatment and management of patients with diabetes through digital healthcare became an issue. In the future, it will be necessary to develop a well-designed telemedicine system to overcome the barriers of time and space and manage diabetes. In particular, it is necessary to develop telemedicine devices for future use in the medical industry by making good use of IT technology and smartphones in South Korea. Furthermore, the risk of chronic diseases, including diabetes, has increased because of the quarantine policy after COVID-19, which could result in a national burden over several years, thereby requiring countermeasures. Furthermore, during the COVID-19 period, obesity and diabetes management in children and adolescents are emerging as an important keywords, implying that long-term diabetes management for children and adolescents is required.

The limitation of this study Is that the research was limited to the period of the new epidemic, and we analyzed only social issues and articles related to diabetes in South Korea. This study does not directly reflect the major concerns or difficulties of diabetic patients during the COVID-19 pandemic. However, this study is meaningful because it identified issues related to chronic diseases that may cause difficulties during the acute epidemic of infectious diseases. Second, publication bias may be present because news outlets tend to publish articles that would have a higher level of public interest. Moreover, the quality of information in news articles varies, ranging from public opinions to in-depth research studies. This study is significant because it uses big data related to diabetes that was reported in the mass media during the new epidemic to identify problems in the health management of patients with diabetes during a new epidemic and discuss areas that should be considered for future interventions. Moreover, it is significant that this occurrence provided an opportunity to identify social problems related to patients with diabetes by categorizing diabetes-related issues during the COVID-19 pandemic using the analysis of “topic modeling,” which has recently been actively used in various academic fields. In future studies, analysis from multiple perspectives is suggested through time-series analytical techniques such as extension of the data collection period and dynamic topic modeling.

## Figures and Tables

**Figure 1 healthcare-11-00957-f001:**
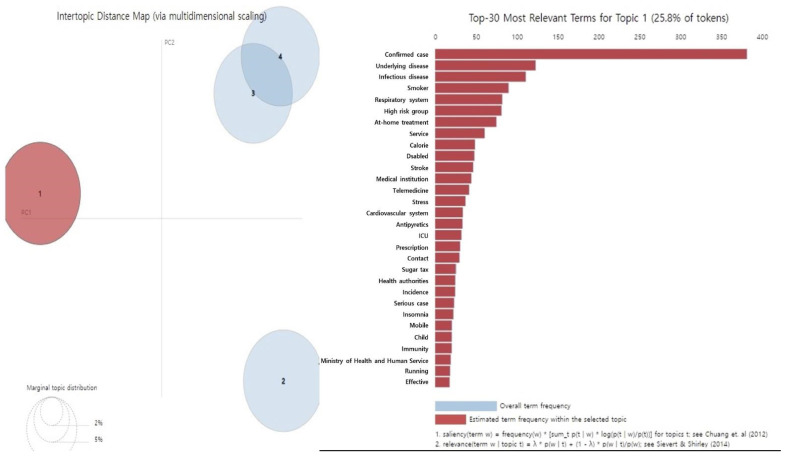
Results of LDA topic modeling ‘Topic 1’ in people with diabetes during the COVID-19 [12,13].

**Figure 2 healthcare-11-00957-f002:**
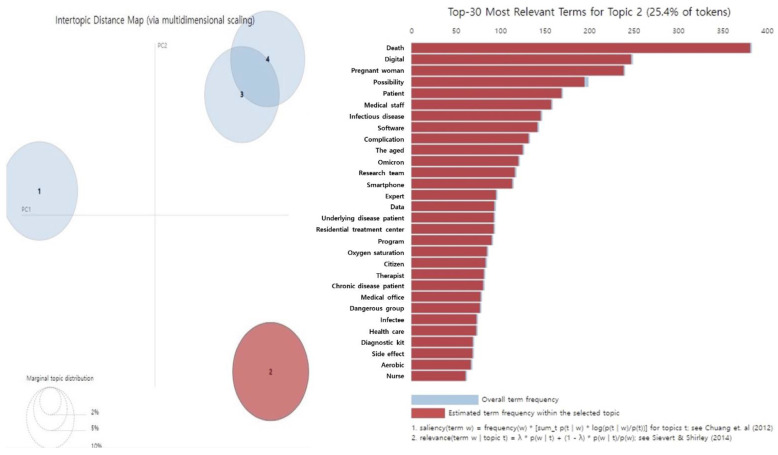
Results of LDA topic modeling ‘Topic 2’ in people with diabetes during the COVID-19 [12,13].

**Figure 3 healthcare-11-00957-f003:**
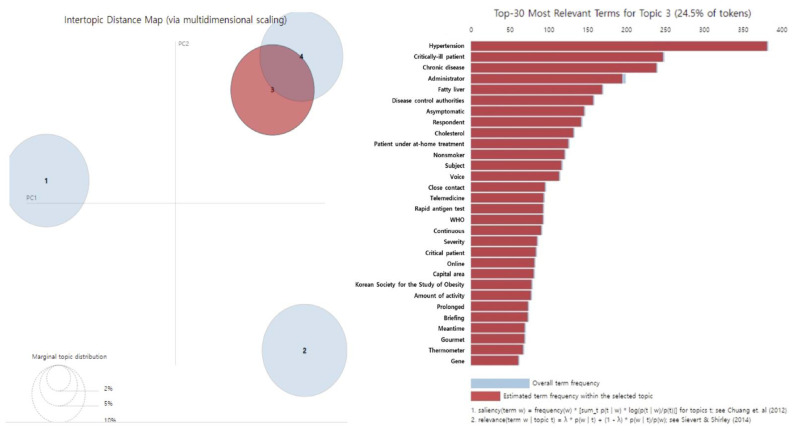
Results of LDA topic modeling ‘Topic 3’ in people with diabetes during the COVID-19 [12,13].

**Figure 4 healthcare-11-00957-f004:**
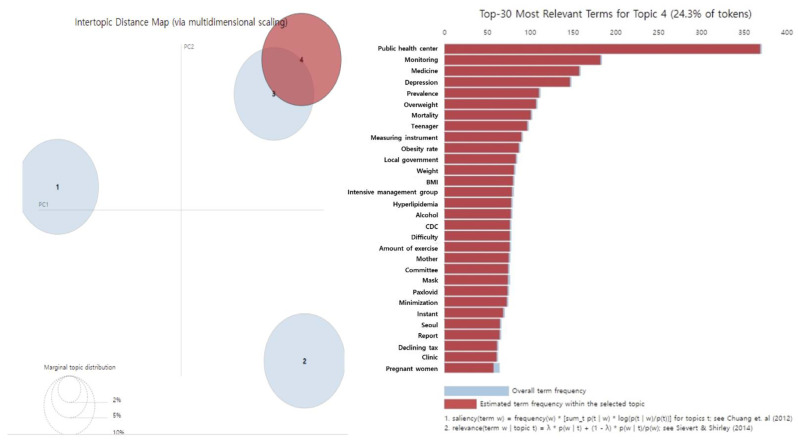
Results of LDA topic modeling ‘Topic 4’ in people with diabetes during the COVID-19 [12,13].

**Table 1 healthcare-11-00957-t001:** Frequency To Rank of 25 Keywords.

Rank	Term	TF ^1^	TF-IDF ^2^
1	COVID-19	539	0.96
2	Chronic disease	440	0.58
3	Health	209	0.27
4	Vaccine	186	0.78
5	Hospital	178	1.31
6	Medical treatment	161	0.57
7	Management	158	0.65
8	Infection	138	0.69
9	Diabetes mellitus	132	0.59
10	Telemedicine	95	0.88
11	Exercise	85	0.74
12	Human	80	0.94
13	Government	79	0.28
14	Digital	78	0.87
15	Confirmed cases	76	1.09
16	Symptom	74	0.12
17	Obesity	74	0.63
18	Care	73	1.12
19	Society	58	1.08
20	Risk	57	0.65
21	Social isolation	53	1.38
22	Service	53	0.90
23	Support	51	1.04
24	Sequelae	51	0.34
25	Death	49	0.72

^1^ TF: Term Frequency. ^2^ TF-IDF: Term Frequency–Inverse Document Frequency.

**Table 2 healthcare-11-00957-t002:** Results of LDA Topic Modeling on People with diabetes during COVID-19.

Rank	Topic 1	Topic 2	Topic 3	Topic 4
1	Confirmed case	Death	Hypertension	Public health center
2	Underlying disease	Digital	Critically-ill patient	Monitoring
3	Infectious disease	Pregnant woman	Chronic disease	Medicine
4	Smoker	Possibility	Administrator	Depression
5	Respiratory system	Patient	Fatty liver	Prevalence
6	High risk group	Medical staff	Disease control authorities	Overweight
7	Being treated for COVID-19 at home	Infectious disease	Asymptomatic	Mortality
8	Service	Software	Respondent	Teenager
9	Calorie	Complication	Cholesterol	Measuring instrument
10	Disabled	The aged	Patient under being treated for COVID-19 at home	Obesity rate
11	Stroke	Omicron	Nonsmoker	Local government
12	Medical institution	Research team	Subject	Weight
13	Telemedicine	Smartphone	Voice	BMI
14	Stress	Expert	Close contact	Intensive management group
15	Cardiovascular system	Data	Telemedicine	Hyperlipidemia
16	Antipyretics	Underlyimg disease patient	Rapid antigen test	Alcohol
17	ICU	Residential treatment center	WHO	CDC
18	Prescription	Program	Continuous	Difficulty
19	Contact	Oxygen saturation	Severity	Amount of exercise
20	Sugar tax	Citizen	Critical patient	Mother

## Data Availability

The datasets used and analyzed during the current study are available from the corresponding author on request.

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
