# Peer review of "Topic Modeling-Based Analysis of News Keywords Related to Patients with Diabetes during the COVID-19 Pandemic"

_healthcare, 2023, doi:10.3390/healthcare11070957_

Round 1

Reviewer 1 Report

The Authors of the article „Topic Modeling-Based Analysis of News Keywords Related to Patients with Diabetes During the COVID-19 Pandemic” presented important research results, focusing on patients with diabetes. The areas that should be taken into account in planning future interventions have been discussed in detail and sufficiently. In addition, it should be noted that the Authors used a large set of data for the analyses, which guarantees the reliability of the analyses.

Below I post my suggestion:

In my opinion, the results lack  references to literature, for example in the excerpt: „several articles reported that the high risk of infection or death from chronic diseases, including diabetes, was associated with cholesterolemia and fatty liver. Various articles reported that lifestyle changes, such as eating and drinking alone at night, following social distancing, or using home treatment because of COVID-19, could cause fatty liver or elevated cholesterol, thereby increasingthe risk of metabolic diseases such as obesity and diabetes”. The same remark applies to the other passages chapters Results.

Author Response

Reviewer 1

Comment 1:

In my opinion, the results lack references to literature, for example in the excerpt: „several articles reported that the high risk of infection or death from chronic diseases, including diabetes, was associated with cholesterolemia and fatty liver. Various articles reported that lifestyle changes, such as eating and drinking alone at night, following social distancing, or using home treatment because of COVID-19, could cause fatty liver or elevated cholesterol, thereby increasingthe risk of metabolic diseases such as obesity and diabetes”. The same remark applies to the other passages chapters Results.

Response 1: We have revised contents or references according to your comments.

Reviewer 2 Report

This is a very nice study conducted using a novel and underused methodology. Findings are interesting too. I only have some minor comments to make. 

1. Present the strengths and limitations of the study. 

2. Reduce the text of Conclusion - just highlight the key findings and their implications. 

3. Abstract: Conclusion is vague. Be specific on what implications the study will have. 

4. Avoid using the term "diabetics" but say "people with diabetes".

5. Why did the authors restrict their search to newspapers and not social media like Twitter and Facebook?

Author Response

Comment 1: This is a very nice study conducted using a novel and underused methodology. Findings are interesting too. I only have some minor comments to make. Present the strengths and limitations of the study.

Response 1: We have revised contents according to your comments.

Comment 2. Reduce the text of Conclusion - just highlight the key findings and their implications.

Response 2: We have revised contents according to your comments.

Comment 3. Abstract: Conclusion is vague. Be specific on what implications the study will have.

Response 3: We have revised contents according to your comments.

Comment 4. Avoid using the term "diabetics" but say "people with diabetes".

Response 4: We have revised contents according to your comments.

Comment 5. Why did the authors restrict their search to newspapers and not social media like Twitter and Facebook?

Response 5: In South Korea, many media companies were publishing articles during the COVID-19 pandemic, and Twitter and Facebook can bring data, but it was difficult to obtain consent from a large number of unspecified users for research data because they contain personal information. We have reflected your opinions and have inserted the relevant information into the text.

Reviewer 3 Report

Comment 1:   Line 64-65.  The sentence isn’t accurate.  Topic modeling analysis cannot address “how media reports affect the management of patients with diabetes….”  as the purpose of topic modeling is to describe and identify both obvious and not as obvious patterns.  I would suggest deleting this sentence completely. I would reframe the objective closer to what was stated in the abstract (lines 19-20) “describe the health management of diabetes during the COVID-19 pandemic to inform future diabetes management strategies.  

Comment 2:  Methods: It is not entirely clear to me if all the articles were in English or not?  In lines 103-104, it stated that the structural SVM algorithm ….can recognize and classify Korean predicates with a performance of 97.13%.   Furthermore lines 112-114 stated that in this study, keywords extracted from characters other than Korean characters, such as CHINESE characters, numbers, and English.  It is clear that English articles were include and English keywords were used, but were Korean language articles also included?  Please clarify.

Comment 3: Methods: line 85. The 226 articles that were included, were they unique articles? Multiple news media outlets will often publish identical content.

Comment 4: Results lines 158-160. The authors stated that TF-IDF based analysis showed words related to social isolation, care, and support were primarily associated with Covid-19 and diabetes.  Was this a separate analysis not shown in Table 1?   According to Table 1, the relevance of Diabetes mellitus is on the lower end relevance based on TF-IDF with 0.59.  This points that the while Diabetes is a common term in articles, it also appears in many articles.  The other terms mentioned have a much higher TF-IDF, indicating that they did not appear in as many articles.

Comment 5: Figure 2 is showing results based on Topic 3; Figure 3 is showing results based on Topic 2.  Heading/footnote also mislabeled.

Comment 6: What is at home treatment? Is this referring to telemedicine? Being treated for COVID at home? Or simply taking medication for chronic conditions such as diabetes?

Comment 7:  The authors should address in the results how Topic 3 and Topic 4 have more overlapping dimensions, in particular, obesity.

Comment 8: Discussion: Unsurprisingly, there are overlaps between all 4 topics, which should be mentioned in the discussion. This overlap is also a limitation for topic modelling.   The authors could consider restructuring the discussion based on themes as opposed to the 4 overlapping topics to avoid repetitiveness and maintain consistency with results.   I can see three themes that could be discussed. 1) Diabetes is one of numerous co-morbidity that makes an individual more vulnerable to severe COVID-19 outcomes. Moreover, confirmed COVID-19 patients could also be at an increased risk for developing diabetes.  2) During COVID-19, patients were fearful of contracting COVID-19 during in-person clinic/hospital visits, leading to poor or delayed disease management.  Telemedicine is a good alternative for managing diseases. 3) Social isolation led to changes in health behavior, such as physical inactivity, contributing to weight gain and obesity.  Obesity exacerbates risk of metabolic syndrome and increases the odds of having a severe outcome from COVID-19.   

Comment 9: Also, should mention that preventing childhood obesity is extremely important as it plays a protective role in preventing the onset of type-two diabetes as adults.

Comment 10:  Another limitation that needs to be mentioned is publication bias.  First, news outlets are selective in what they print.  They tend to print news articles that are more likely to generate public interest and they are tailored to the general public, meaning that provided information is high-level.   Moreover, the quality of information is also not consistent, as information can come from opinions (either from the public or experts), surveys (which can have its own set of limitations) and research studies.

Author Response

Comment 1: Line 64-65. The sentence isn’t accurate. Topic modeling analysis cannot address “how media reports affect the management of patients with diabetes….” as the purpose of topic modeling is to describe and identify both obvious and not as obvious patterns. I would suggest deleting this sentence completely. I would reframe the objective closer to what was stated in the abstract (lines 19-20) “describe the health management of diabetes during the COVID-19 pandemic to inform future diabetes management strategies.

Response 1: We have reflected your opinions and have inserted the relevant information into the text.

Comment 2: Methods: It is not entirely clear to me if all the articles were in English or not? In lines 103-104, it stated that the structural SVM algorithm ….can recognize and classify Korean predicates with a performance of 97.13%. Furthermore lines 112-114 stated that in this study, keywords extracted from characters other than Korean characters, such as CHINESE characters, numbers, and English. It is clear that English articles were include and English keywords were used, but were Korean language articles also included? Please clarify.

Response 2: We have revised contents according to your comments.

Comment 3: Methods: line 85. The 226 articles that were included, were they unique articles? Multiple news media outlets will often publish identical content.

Response 3: We have revised contents according to your comments.

Comment 4: Results lines 158-160. The authors stated that TF-IDF based analysis showed words related to social isolation, care, and support were primarily associated with Covid-19 and diabetes. Was this a separate analysis not shown in Table 1? According to Table 1, the relevance of Diabetes mellitus is on the lower end relevance based on TF-IDF with 0.59. This points that the while Diabetes is a common term in articles, it also appears in many articles. The other terms mentioned have a much higher TF-IDF, indicating that they did not appear in as many articles.

Response 4: We have checked the contents and revised the part where errors occurred during the translation process.

Comment 5: Figure 2 is showing results based on Topic 3; Figure 3 is showing results based on Topic 2. Heading/footnote also mislabeled.

Response 5: We have checked what you pointed out, and the number of figures per topic is correct. However, it was confirmed that there was an error in the title part in the statistical program, and corrections were made.

Comment 6: What is at home treatment? Is this referring to telemedicine? Being treated for COVID at home? Or simply taking medication for chronic conditions such as diabetes?

Response 6: We have revised contents according to your comments.

Comment 7: The authors should address in the results how Topic 3 and Topic 4 have more overlapping dimensions, in particular, obesity.

Response 7: We have revised below

Topic 3 and Topic 4 had many overlapping parts, but the commonality of the two topics was that social isolation led to changes in health behavior, such as physical inactivity, contributing to weight gain and obesity. Obesity exacerbates the risk of metabolic syndrome and increases the odds of having a severe outcome from COVID-19. However, the biggest difference between Topic 3 and Topic 4 was that the subjects of Topic 3 were primarily adults, and the subjects of Topic 4 were mostly children and adolescents. Therefore, to prevent and manage diabetes, appropriate policies and interventions are needed depending on the subject. Particularly, preventing childhood obesity is extremely important as it plays a protective role in preventing the onset of type-two diabetes as adults.

Comment 8: Discussion: Unsurprisingly, there are overlaps between all 4 topics, which should be mentioned in the discussion. This overlap is also a limitation for topic modelling. The authors could consider restructuring the discussion based on themes as opposed to the 4 overlapping topics to avoid repetitiveness and maintain consistency with results. I can see three themes that could be discussed. 1) Diabetes is one of numerous co-morbidity that makes an individual more vulnerable to severe COVID-19 outcomes. Moreover, confirmed COVID-19 patients could also be at an increased risk for developing diabetes. 2) During COVID-19, patients were fearful of contracting COVID-19 during in-person clinic/hospital visits, leading to poor or delayed disease management. Telemedicine is a good alternative for managing diseases. 3) Social isolation led to changes in health behavior, such as physical inactivity, contributing to weight gain and obesity. Obesity exacerbates risk of metabolic syndrome and increases the odds of having a severe outcome from COVID-19.

Response 8: We have added contents according to your comments to the discussion section.

Comment 9: Also, should mention that preventing childhood obesity is extremely important as it plays a protective role in preventing the onset of type-two diabetes as adults.

Response 9: We have added contents according to your comments to the discussion section.

Comment 10: Another limitation that needs to be mentioned is publication bias. First, news outlets are selective in what they print. They tend to print news articles that are more likely to generate public interest and they are tailored to the general public, meaning that provided information is high-level. Moreover, the quality of information is also not consistent, as information can come from opinions (either from the public or experts), surveys (which can have its own set of limitations) and research studies.

Response 10: We have added contents according to your comments to the conclusion section.

Round 2

Reviewer 3 Report

Comment 1: Methods (lines 79-82): There are quite a few published articles that used twitter data. It would be unrealistic to get consent from every user.  Topic modeling should consider only post data and not username or other user information. Plus, usernames will have lower TF-IDF scores and parameters can be set to filter it out.  The authors should remove this add in sentence as it is not a good explanation for why twitter and Facebook data were not used. Since each data source has its own set of limitations, it’s not an issue to only use one source instead of all available sources.  If it’s of interest to the authors in the future, twitter data could be something to consider separately and compared to news source.  

Comment 2:  Thanks for clarifying that everything published is in Korean.

Comment 3: The images for figure 1 (Topic 2 and 3) are very blurry, please provide clearer images.

Comment 4:  Thanks to the authors for being responsive to comments.  I wasn’t requiring the authors to use the exact wording from my comments in the manuscript.   That being said, I would suggest shortening the second limitation section (lines 412-417), as I was addressing the authors vs. a wider audience in my previous comments. I would shorten it to something like….   “ Second, publication bias may be present because news outlets tend to publish articles that would have a higher level of public interest. Moreover, the quality of information in news articles varies, ranging from public opinions to in-depth research studies.”

Author Response

Comment 1:

: Methods (lines 79-82): There are quite a few published articles that used twitter data.It would be unrealistic to get consent from every user. Topic modeling should consider only postdata and not username or other user information. Plus, usernames will have lower TF-IDF scoresand parameters can be set to filter it out. The authors should remove this add in sentence as it isnot a good explanation for why twitter and Facebook data were not used. Since each data sourcehas its own set of limitations, it’s not an issue to only use one source instead of all availablesources. If it’s of interest to the authors in the future, twitter data could be something to considerseparately and compared to news source.

Response 1: We have deleted lines 79-82 according to your comments.

Comment 2: Thanks for clarifying that everything published is in Korean.

Response 2: Thank you.

Comment 3: The images for figure 1 (Topic 2 and 3) are very blurry, please provide clearer images.

Response 3: We have changed Figure 1(Topic 2 and 3) according to your comments.

Comment 4: Thanks to the authors for being responsive to comments. I wasn’t requiring theauthors to use the exact wording from my comments in the manuscript. That being said, I wouldsuggest shortening the second limitation section (lines 412-417), as I was addressing the authorsvs. a wider audience in my previous comments. I would shorten it to something like…. “ Second,publication bias may be present because news outlets tend to publish articles that would have ahigher level of public interest. Moreover, the quality of information in news articles varies, rangingfrom public opinions to in-depth research studies.”

Response 4: We have revised the second limitation section (lines 412-417) according to your comments.